# The Relation between Green Visual Index and Visual Comfort in Qingdao Coastal Streets

**Dong Sun** [1], **Xiang Ji** [1], **Weijun Gao** [2,3,*], **Fujian Zhou** [1], **Yiqing Yu** [1], **Yumeng Meng** [4], **Meiqi Yang** [4], **Junjie Lin** [1] and **Mei Lyu** [4,*]

1. School of Architecture and Urban Planning, Shenyang Jianzhu University, Shenyang 110168, China
2. Innovation Institute for Sustainable Maritime Architecture Research and Technology, Qingdao University of Technology, Qingdao 266033, China
3. Faculty of Environmental Engineering, The University of Kitakyushu, Kitakyushu 8080135, Japan
4. School of Design and Art, Shenyang Jianzhu University, Shenyang 110168, China
* Correspondence: gaoweijun@me.com (W.G.); lynmei@sjzu.edu.cn (M.L.)

**Abstract:** The public's mental health is obviously impacted by the perception of green quantity in urban streets. As one of the important urban spatial indicators, the Green View Index (GVI) reflects the green quantity of streets, which is helpful in revealing the level of street vegetation from the perspective of pedestrians. The GVI can improve the attraction and the visual experience in urban streets. Taking Qingdao Coastal Streets as an example, the study used OpenStreetMap, Baidu Street View (BSV) image, DeepLabV3+ semantic segmentation, and the SD method to obtain the GVI and Visual Comfort (VICO), and the correlation and influence mechanisms were discussed. The result showed that the greening landscape of the overall Qingdao Coastal Streets was of high quality, and the historic district was the most outstanding. The greening quality was a little low in the transitional district and the western modern district, which should be improved. In addition, the relationship between GVI and VICO showed a strong positive correlation. The spatial distribution of the VICO was more consistent with the GVI. The street VICO was affected by the GVI, plant richness, the street scale, and landscape diversity. Moreover, with the increase of the GVI, the increase trend of the VICO instead gradually decreased. The contribution of this study was not only accurately diagnosing the problems of street greening quality, shedding light on the relationship between GVI and VICO, but also providing theoretical support for urban greening planning and management, especially for healthy street design.

**Keywords:** coastal streets; street view image; semantic segmentation; green view index (GVI); visual comfort (VICO)

## 1. Introduction

Urban green spaces have been regarded as a crucial landscape design factor in urban environments [1,2]. The shifts in vegetation cover and composition in street space can significantly affect the urban environment [3].

In the 1970s, Kaplan R. proposed the visual landscape theory [4]. Visual perception is the most important sense, which can obtain more than 80% of its information from the environment surrounding it [5–7]. In the "visual landscape", the subject is human and the object is the landscape elements. The form and quantity of the elements have a significant impact on landscape perception [8,9]. The evaluation process of visual landscape quality is also a process of comparative analysis of the aesthetic perception of different landscape spaces [6,10]. "Aesthetic and Affective Response Model" has combined evolutionary aesthetics with affection, emphasizing the initial emotional response of the "like–dislike" for landscape aesthetics [11]. The landscape visual evaluation includes five models: ecological model, formal aesthetic model, psychophysical model, psychological and phenomenological model [10,12,13]. Ecological quality can be reflected in the visual

perception immediately, and the visual comfort (VICO) is largely responsible for the overall ecological comfort grade. In the early research, the visual landscape focused on undeveloped natural areas. With the urbanization development, the research object turned to urban and rural areas in recent years. In general, landscape visual environment evaluation has been applied to natural landscapes, urban and rural landscapes, and ecological landscapes, as well as playing an indispensable role in landscape improvement, ecological restoration, and regional governance [14].

In urban environments, street greenery provides environmental, economic, and social services to cities [15], such as pollution reduction [16,17], noise elimination [18], urban biodiversity [19–21], and aesthetic [22,23]. In addition, urban street greenery plays an important role in public health [24–26]. Former studies indicate the amount, visibility, and accessibility of urban greenery have a positive correlation with the physical, emotional, and mental health of the public [27–31]. It could promote psychological well-being [32–34] mental restoration, stress reduction, and emotional health [35–38], as well as the general public health of urban residents [39–43]. Researchers found that VICO was one of the most significant functions of urban greenery [44]. Urban street greenery is an important element that has a direct impact on the visual evaluation and perception of space [45–48] and also improves landscape aesthetic evaluation [2,49–51].

The urban coastal street is the region with the most diverse natural environment in the city, including ecological landscape, artificial landscape, and social and cultural landscape [52,53]. Moreover, the coastal street is a key area for tourism and leisure, and it can show urban styles and features. At present, the purpose of urban regeneration in coastal streets is to combine urban economy and urban development to build comfortable artificial landscapes and sustainable ecological environments [54,55]. As the crucial landscape element in the urban green system, urban street greenery makes a very important contribution to attractiveness and comfort [25,48,56].

In previous studies, indicators such as canopy cover [57], leaf area index [58], total leaf biomass [59], leaf area density [60], green plot ratio [61], percentage of green space coverage, and urban green space per capita [44] have been used to objectively measure the quantity and morphology of urban green plants. With the expansion of the study scale from the neighborhood level to the city level, more quantitative indicators are developed such as floor green view (FGV) [62], street greenery [63], green view [64], green view index (GVI) [65], and green space ratio (GSR) [66]. Among them, the concept of GVI was formally presented by Japanese professor Yoji Aoki in 1987. He found that most people have a favorable impression of street landscape environments with more than 30% GVI [44,67]. And then a Japanese environmental geographer, Takazo Ohno, proposed the theory of GVI, which is widely accepted internationally.

GVI has been widely used to study the spatial distribution and equity of urban green spaces [68], which also provides the possibility for quantitative analysis of the relation between physical features and space quality [69]. Researchers used the GVI to evaluate the ecological service function [70] and public visual ecological quality of urban greenery [71]. The visibility of greenery and the GVI distribution could affect human perceptions of safety [72,73]. It has been demonstrated that GVI is more closely connected with physical activity and health [25,72,74,75]. Ye has analyzed the correlation between the street GVI and vegetation cover in Shanghai [76,77]. Xiao has focused on the influence of street GVI on mental health [78,79]. Xu et al. have studied the attraction of the GVI and found that the GVI had an impact on public perception [80,81]. Long et al. have compared the GVI between some cities [68,82]. Cui has analyzed the GVI differences among different road grades [83,84]. Yang has revealed the relationship between street physical features and VICO [78,85].

The main methods of measurement contained photographic interpretation, field survey, and remote sensing images in early GVI studies. The method, which combines field surveys and photography interpretation, is the most commonly used to evaluate the visibility of urban greenery [64,86]. The researchers took pictures at random sample points

along streets, which were obtained based on the pedestrian's perception; four photos were taken in four directions (north, south, east, and west) to evaluate the street greenery at each sample point [64]. However, the method was less efficient in acquiring and processing data, and the study scope was limited to small areas [87].

With the wide application of Street View Image big data, the limitations of sample photo collection in terms of sample size, time, and area were solved [88]. Street View Image has been proven to be an effective and reliable tool for measuring built environments on various scales, such as streets and neighborhoods [89–91]. Recently, Street View Image has not only been used to measure eye-level visibility of green vegetation [67,92] but has also been the subject of various research in the field of street space [48,65,79,90,93]. Based on Street View images, scholars have found many methods to measure GVI. Tang and Hao obtained the massive street view image through Tencent Map and extracted the proportion of green areas at 60°–180° chromatogram by MATLAB [64]; Li et al. evaluated the street greenery manually exported through pixel-based color recognition in Photoshop [24].

Computer vision technology such as semantic segmentation has provided possibilities for dealing with large-scale data [67,94,95]. Semantic segmentation is a deep learning algorithm that associates the labels or categories with each pixel in the images. It uses SegNet [96–98], PSPNet [99], Cityscapes, and other datasets to classify and quantify the landscape elements such as plants, buildings, sky, and green spaces in photos to identify and form a set of pixels with different categories. Combining Street View big data and semantic segmentation techniques, many scholars have conducted large-scale and fine-grained urban GVI studies [62–64].

Some methods are often used for the evaluation of urban street plant landscapes, such as the analytic hierarchy process [100,101], factor analysis, the semantic difference method, physiological and psychological indicators [102], and scenic beauty estimation (SBE). And the evaluation models are constructed using expert evaluation methods, public aesthetic preferences, or a combination of qualitative and quantitative methods [103]. Zhao and Li (2014) have used the SBE method to evaluate the street tree landscape and analyze the differences among street tree communities [104]. Yan et al. (2011) have used the BIB-LCJ method to evaluate street tree communities, analyze public preferences, and propose optimization strategies [105].

The semantic differential (SD) method is the integration of subjective judgment and objective feature perception [106,107]. The public's perceptual information, attitudes, and preferences of VICO in street space were collected based on opposite adjective pairs [28,108,109]. Compared with other evaluation methods such as the Likert scale, the SD method is relatively reliable and valid. Nowadays, many studies have used the SD method, and it is usually combined with other statistical methods [110,111]. Zhao et al. (2022) have used the semantic difference method (SD) and factor analysis method to construct the visual evaluation system for the street landscapes in historical cultural districts [112]. Shao and Liu have used the psychophysical method to analyze the relationship between street visual quality and spatial elements [113,114]. The landscape visual quality can be quantified by spatial measurement, such as volume, area, landscape diversity, shape, connectivity, color, topography, and openness [115]. Han and Dong have selected seven physical indicators—visual entropy, color diversity index, skyline index, street width, building height, and sky openness index—to evaluate the street visual quality [116].

Nowadays, most scholars respectively study street physical features and people's psychological perception of street. Few people discuss the influence mechanism between street physical features and psychological perception. Therefore, Qingdao Coastal Streets were chosen as the study site in the study. Based on the street view image data and Semantic Segmentation technology, the GVI was objectively measured, while the expert evaluation for VICO was carried out by the SD method. The relationship and mechanism between the GVI and the public VICO were analyzed. The study results contribute to improving the landscape quality and healthy environment in Qingdao Coastal Streets. It will promote urban regeneration and the economic development of Qingdao.

## 2. Materials and Methods

### 2.1. Study Site and Data

As a typical coastal city in the north of China, Qingdao is surrounded by the sea on three sides, with the pattern of a mountain-sea city. The landscape resources, both natural and cultural, are rich. The study area is located in the southern coastal area of Qingdao (Figure 1), from Xilingxia Road in the west to Donghai Middle Road in the east, with a total length of 17.15 km. Qingdao Coastal Streets are an important tourist area. Street greening is the most important landscape element with high research value in Qingdao Coastal Streets. The most critical issues in the study of Qingdao Coastal Street are as follows: (1) efficiently and accurately quantify the analysis of the GVI and VICO in coastal streets; (2) a comparative analysis of the spatial heterogeneity of the GVI and VICO in various types of districts; and (3) study the relationship and influence mechanisms between the GVI and VICO.

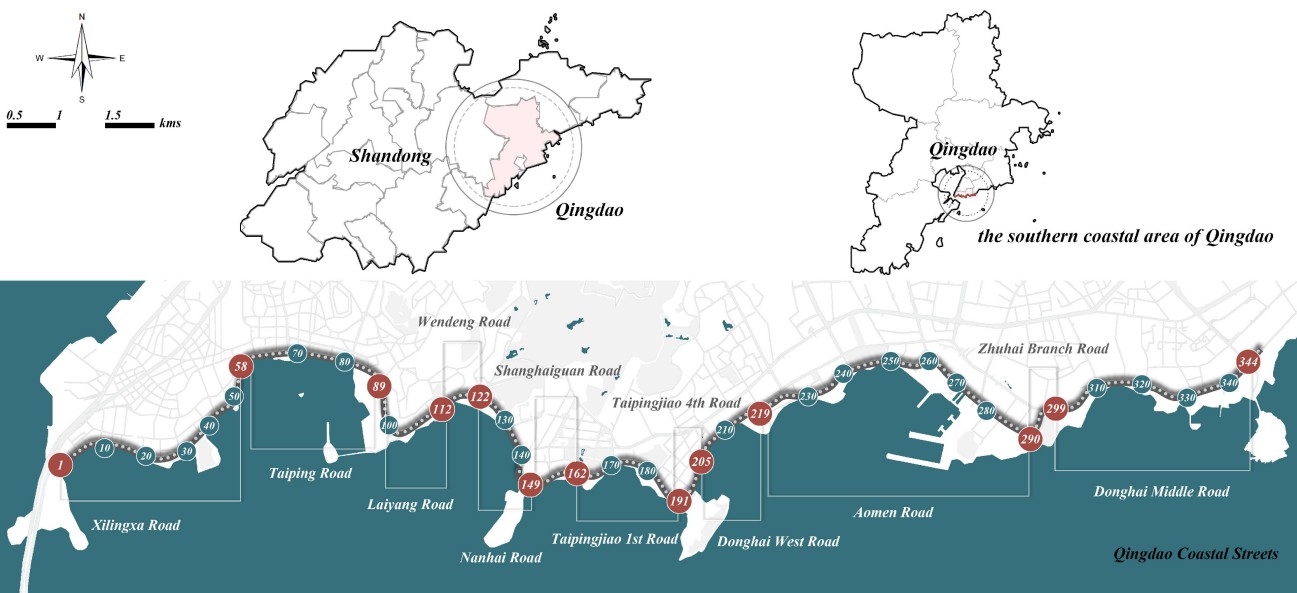

**Figure 1.** Study site and distribution of sample points.

Figure 2 shows the data collection, processing, and analysis process for this study. In the study, Baidu Street View (BSV) images were used as the main data source. Baidu Map is the biggest streetscape data provider in China. The users can make corresponding download requests to Baidu Map (https://map.baidu.com/, accessed on 30 March 2022) by setting different API parameters, so as to obtain the street view image needed for research. Based on OpenStreetMap, the vector road network data of Qingdao Coastal Streets was obtained and imported into ArcGIS10.3 to delete and simplify the road network. With the help of "Split line At Vertices" (one of the edit functions) and the "Feature Vertices to Points" tool, the street sampling points with a distance interval of 50 m were generated. And the latitude and longitude coordinates of the sampling points were obtained with the help of the "Add X Y Coordinates" tool. After verification, 344 coastal street sample points were obtained. In order to better simulate the actual pedestrian perspective [117], the following parameters for the BSV image were set: size: 960 × 720; pitch: 0 degree; heading: 0, 60, 120, 180, 240, and 300 degrees; location: Baidu longitude and latitude coordinates of the sample point converted by the coordinate conversion program. Through the automatic image crawling program written in Python, BSV images of 8 angles of each sample point were obtained (Figure 3). The street view images were taken from May to July 2015–2020. A total of 2752 valid BSV images were obtained.

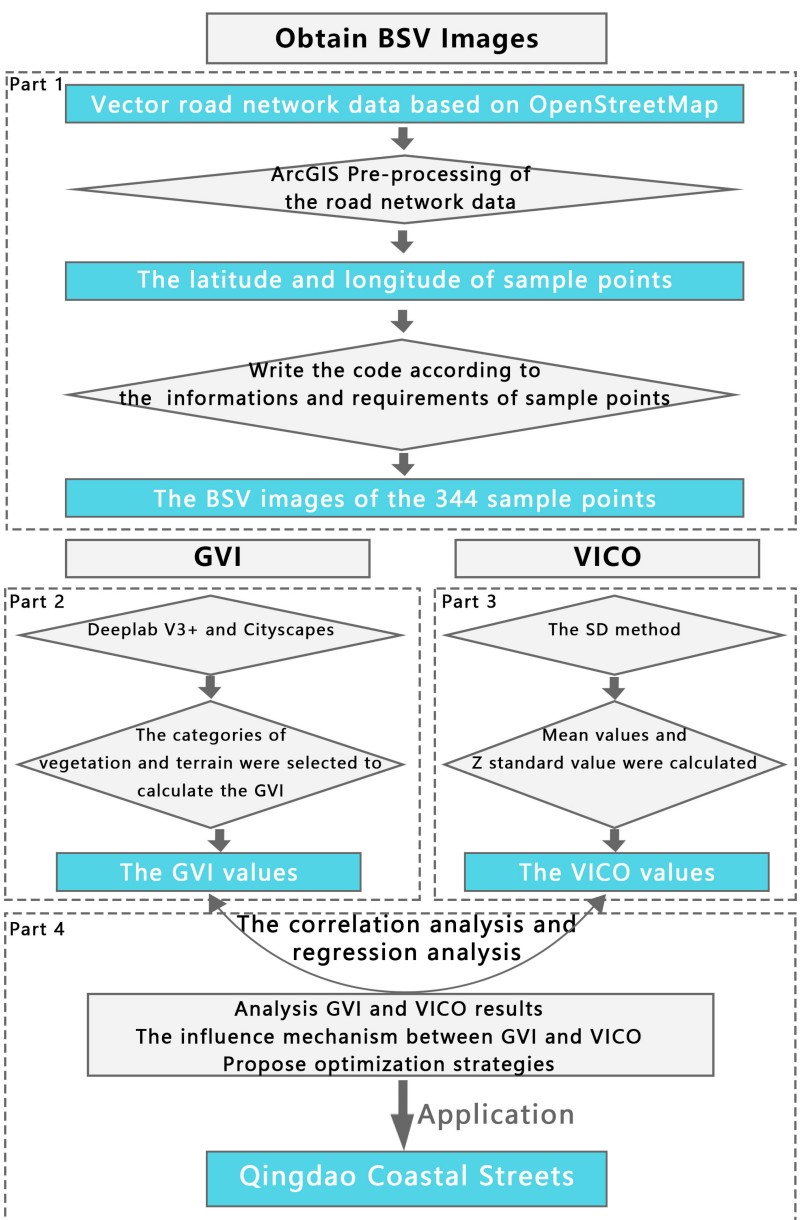

**Figure 2.** Methodology and application step diagram.

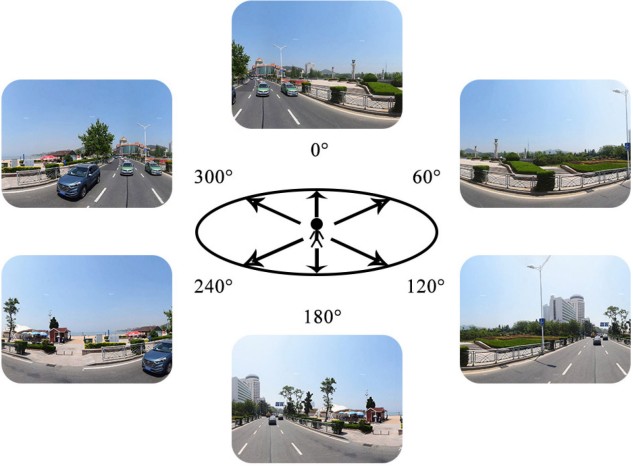

**Figure 3.** Collection of BSV images of sample points.

## 2.2. Calculation of Green View Index (GVI) of Street View by DeepLab V3+

A Japanese scholar [69] first put forward the concept of GVI to analyze green quantity. GVI is an indicator of the visibility of greenery at street level. And it is the pixel proportion of the green vegetation in the street view image.

There are three methods for getting the street view GVI: (1) obtaining green plant pixel values in the histogram by Photoshop software [118] or using GIMP software to divide the image into a $40 \times 40$ grid to roughly calculate the proportion of green plant pixels. (2) Writing code to analyze the composition of different colors in a BSV image, converting the color mode from RGB to HSI or HSV, and marking the pixels with threshold degrees between 75 and 170 or 60 and 180 as green plant pixels [88,94]. (3) Using semantic segmentation techniques such as PSPnet, SegNet, and DeepLab and training datasets such as CityScapes, ADE-20K, and CS to recognize the landscape elements of a street view image [117,119]. The plant elements were extracted to calculate GVI (Figure 4). The first method is time-consuming and labor-intensive. And it is suitable for a small number of street-view images. Rencai Dong [94] compared manual Photoshop segmentation with semantic segmentation and pointed out that the results of both methods are consistent. Aiping Gou [119] compared and analyzed the PSPnet, DeepLabV3+, and HSV for extracting color thresholds. It was found that DeepLabV3+ has high accuracy.

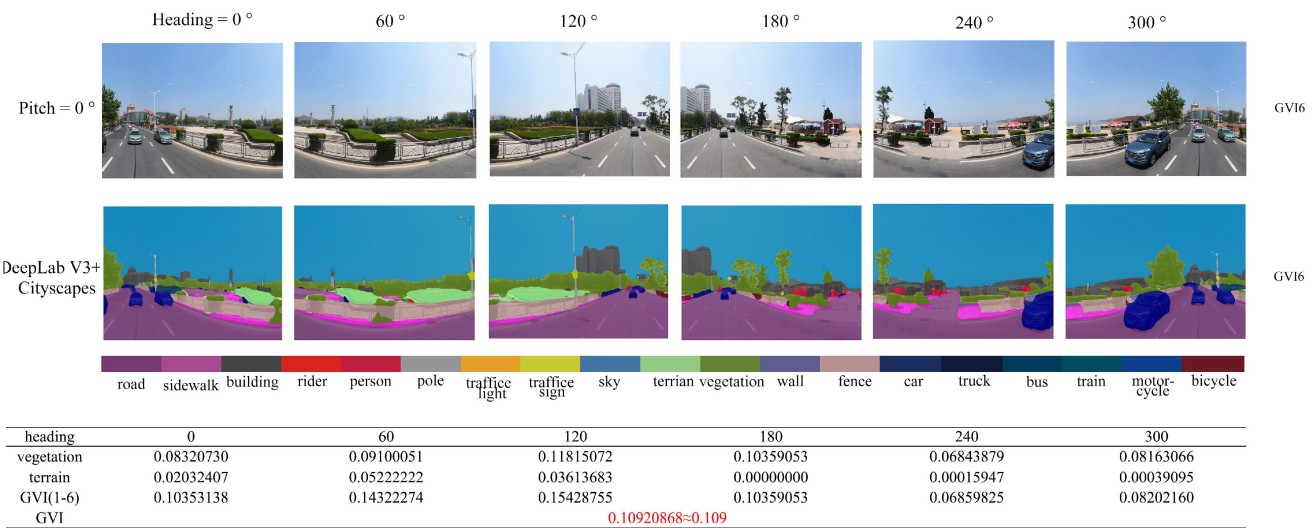

| heading | 0 | 60 | 120 | 180 | 240 | 300 |
|---|---|---|---|---|---|---|
| vegetation | 0.08320730 | 0.09100051 | 0.11815072 | 0.10359053 | 0.06843879 | 0.08163066 |
| terrain | 0.02032407 | 0.05222222 | 0.03613683 | 0.00000000 | 0.00015947 | 0.00039095 |
| GVI(1-6) | 0.10353138 | 0.14322274 | 0.15428755 | 0.10359053 | 0.06859825 | 0.08202160 |
| GVI | | | 0.10920868≈0.109 | | | |

**Figure 4.** The semantic pixel-wise labeling of landscape elements.

Therefore, the DeepLabV3+ and CityScapes datasets were used to identify landscape elements in the BSV image. The CityScapes dataset can identify 19 categories. It is one of the most authoritative and professional evaluation sets at present. The categories of vegetation and terrain were selected to calculate the GVI of the sample points. The formula for calculating GVI is as follows:

$$ \text{GVI} = \frac{1}{n} \sum_{i=1}^{n} V_n + \frac{1}{n} \sum_{i=1}^{n} T_n \{ i \in (1, 2, \ldots, n) \} \tag{1} $$

where $n$ denotes the number of sample point images, which is 6 in this paper; $V_n$ denotes the proportion of vegetation pixels; and $T_n$ denotes the proportion of terrain pixels.

## 2.3. Evaluation of Visual Comfort (VICO) of Street View Based on SD Method

The SD (semantic differential) method proposed by Charles E. Osgood [120] quantitatively describes the concept and structure of the research object by analyzing the established scales. The SD method is widely used in landscape evaluation [121], which usually requires 20–50 professional observers [122,123]. In this study, they invited 30 students and 15 teachers with major backgrounds in architecture, urban planning, and landscape architecture.

After explaining the meaning of VICO and its adjective pair (uncomfortable-comfortable), they were asked to evaluate the BSV image of the sample points on a 5-point scale (1–5). Each image was shown for 20 s, and 45 evaluation forms were retrieved, which were all valid after inspection.

### 2.4. Statistical Analysis

In this study, for discussing the relationship between the physical feature GVI and the perceptual feature VICO in Qingdao Coastal Streets, the GVI and VICO were conducted through Pearson correlation analysis and regression analysis by SPSS 25.0.

## 3. Results

According to the previous research results of Qingdao Coastal Streets [124], Qingdao was divided into three parts, namely, historic urban area, central urban area, and eastern new area. And according to the difference in architectural styles and features, Qingdao Coastal Streets were divided into three types (Figure 5): transitional district (the sample points 1–67), historic district (the sample points 68–206), and modern district (the sample points 207–344). Then, the GVI and VICO in the three types of Qingdao Coastal Streets were analyzed and discussed. Meanwhile, the overall Qingdao Coastal Streets were divided into 12 sections. Then, the GVI and VICO in the three types of Qingdao Coastal Streets were analyzed and discussed. Table A1 shows GVI and VICO values for each sample point.

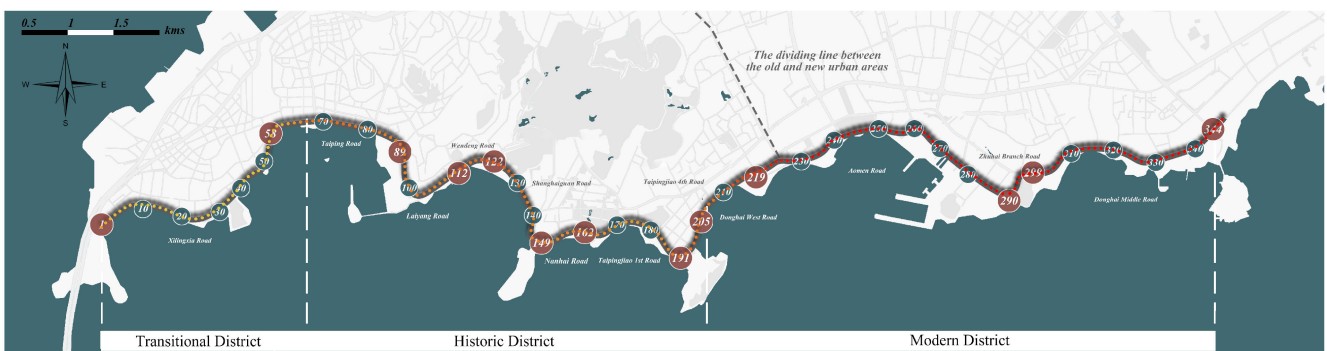

**Figure 5.** Three types of Qingdao Coastal Streets.

### 3.1. The GVI Analysis

Through the analysis of the mean values of GVI in the overall Qingdao Coastal Streets and the three types of districts (Figure 6a), it could be found that the overall GVI in Qingdao Coastal Streets was very high (0.344) and created a strong green perception for the pedestrians. And GVI-historic (0.418) > GVI-modern (0.314) > GVI-transitional (0.251). Among them, the GVI of the historic district was the best, which was higher than the GVI of the overall streets (0.344). The second was the modern district, which was similar to the overall streets. The GVI of the transitional district was the lowest, while the GVI was also higher than 25%. It was a strong green perception grade. In addition, the standard deviation of the three types of districts was close to that of overall Qingdao Coastal Streets. It indicated that the fluctuation range of green quantity was relatively consistent in the three types of districts. In the transitional district, the GVI values of sample points 57–60 were very low (<5%). The GVI values of sample points 38–52 were lower than 15%. The GVI values of sample points 13–37 were generally higher than 35%, and the highest was 0.687, with a very high green quantity. In the historic district, sample point 135 had the lowest GVI (0.091) and sample point 161 had the highest GVI (0.740). In the modern district, sample point 224 had the lowest GVI (0.024) and sample point 287 had the highest GVI (0.811).

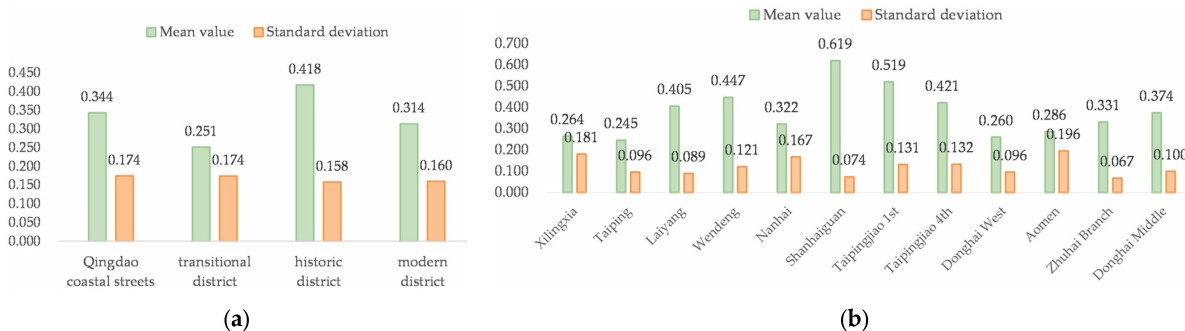

**Figure 6.** (**a**) The descriptive analysis of GVI values in the overall Qingdao Coastal Streets and the three types of districts; (**b**) the descriptive analysis of GVI values in the 12 roads.

Figure 6b shows that: (1) The GVI of Laiyang Road (0.405), Wendeng Road (0.447), Shanhaiguan Road (0.619), Taipingjiao 1st Road (0.519), Taipingjiao 4th Road (0.421) and Donghai Middle Road (0.374) were higher than overall Qingdao Coastal Streets (0.344), and the green quantity was large; the GVI of Xilingxia Road (0.264), Taiping Road (0.245), Nanhai Road (0.322), Donghai West Road (0.260), Aomen Road (0.286), and Zhuhai Branch Road (0.331) was lower than overall Qingdao Coastal Streets (0.344), and the greening needed to be improved. The standard deviations of Xilingxia Road (0.181), Wendeng Road (0.121), Nanhai Road (0.167), Taipingjiao 1st Road (0.131), Taipingjiao 4th Road (0.132), and Aomen Road (0.196) were higher, which indicated that the distribution of green quantity was different among them. Taiping Road (0.096), Laiyang Road (0.089), Shanhaiguan Road (0.074), Donghai West Road (0.096), Zhuhai Branch Road (0.067), and Donghai Middle Road (0.100) had lower standard deviations, which indicated that the green quantity distribution was relatively balanced. The greening was continuous. Among the 12 roads, Shanhaiguan Road had the highest GVI value (0.619) and lowest standard deviation (0.074), which showed that the greening distribution of this road overall was lush and balanced. It provided a valuable reference for the green design; Taiping Road had the lowest GVI value (0.245) and low standard deviation (0.096), which showed that there were fewer plants in all sample points of Taiping Road. The street greening should be improved.

Japanese scholar (Orihara Natsuyuki, 2006) divided the GVI into five grades: 0~5%, 5~15%, 15~25%, 25~35%, and 35% or more. Because in Qingdao Coastal Streets, there were almost no sample points with GVI 0~5%. The GVI was divided into four grades (Table 1): 0~15%, 15~25%, 25~35%, and 35% or more.

**Table 1.** The GVI grades of sample points.

| Grade | 0~15% GVI | 15~25% GVI | 25~35% GVI | 35~100% GVI |
|---|---|---|---|---|
| Sample points | 1, 2, 3, 4, 30, 38, 39, 40, 41, 44, 45, 46, 47, 48, 49, 50, 51, 52, 57, 58, 59, 60, 63, 85, 125, 127, 132, 135, 136, 207, 220, 223, 224, 225, 226, 227, 228, 229, 230, 231, 232, 245, 246, 247, 248, 249, 251, 254, 255, 256, 268 | 5, 6, 7, 8, 9, 10, 11, 12, 24, 32, 42, 54, 55, 56, 61, 62, 64, 65, 66, 73, 76, 86, 87, 88, 89, 95, 99, 120, 124, 126, 131, 133, 134, 137, 175, 210, 211, 212, 213, 217, 233, 235, 241, 244, 250, 252, 253, 257, 259, 262, 263, 270, 271, 280, 294, 309, 312, 315, 330, 336, 344 | 14, 15, 16, 20, 27, 29, 33, 34, 43, 53, 67, 68, 69, 70, 71, 72, 74, 75, 77, 81, 82, 83, 84, 100, 107, 119, 128, 129, 130, 143, 169, 176, 192, 195, 197, 198, 200, 202, 208, 209, 214, 215, 216, 219, 221, 222, 234, 236, 237, 238, 240, 242, 258, 260, 261, 283, 291, 293, 295, 296, 304, 305, 306, 307, 308, 310, 311, 313, 317, 320, 321, 337, 340 | 13, 17, 18, 19, 21, 22, 23, 25, 26, 28, 31, 35, 36, 37, 78, 79, 80, 90, 91, 92, 93, 94, 96, 97, 98, 101, 102, 103, 104, 105, 106, 108, 109, 110, 111, 112, 113, 114, 115, 116, 117, 118, 121, 122, 123, 138, 139, 140, 141, 142, 144, 145, 146, 147, 148, 149, 150, 151, 152, 153, 154, 155, 156, 157, 158, 159, 160, 161, 162, 163, 164, 165, 166, 167, 168, 170, 171, 172, 173, 174, 177, 178, 179, 180, 181, 182, 183, 184, 185, 186, 187, 188, 189, 190, 191, 193, 194, 196, 199, 201, 203, 204, 205, 206, 218, 239, 243, 264, 265, 266, 267, 269, 272, 273, 274, 275, 276, 277, 278, 279, 281, 282, 284, 285, 286, 287, 288, 289, 290, 292, 297, 298, 299, 300, 301, 302, 303, 314, 316, 318, 319, 322, 323, 324, 325, 326, 327, 328, 329, 331, 332, 333, 334, 335, 338, 339, 341, 342, 343 |

Figure 7 shows the spatial distribution of GVI in Qingdao Coastal Streets. It showed that the sample points with GVI higher than 35% were concentrated in the historic district and the eastern modern district. Except for Taiping Road, Nanhai Road, and the western Aomen Road, the GVI of other streets in the historic district and the modern district was higher than 35%. The sample points with 25~35% GVI were concentrated in the middle of Xilingxia Road, Taiping Road, and the Donghai Middle Road. Most of them were close to the sample points, with GVI higher than 35%. The sample points with 15~25% GVI and 0~15% GVI were mostly concentrated in the eastern and western transitional districts and the western modern district.

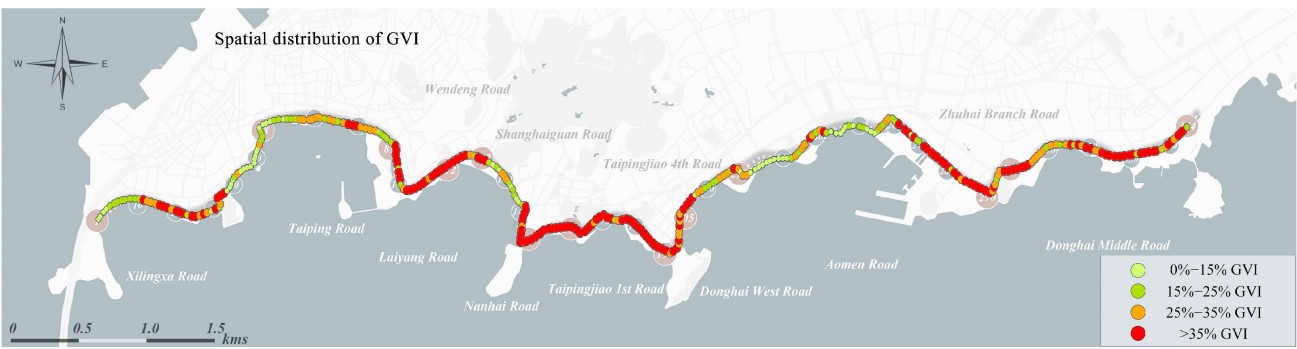

**Figure 7.** Spatial distribution of GVI.

The proportion of each GVI level is shown in Figure 8. As Figure 8a shows: The best green grade (GVI 35–100%) accounted for the highest share in the historic district and modern district, while in the transitional district, the worst green grade (GVI 0–15%) accounted for the highest share, and the sample points with 0~25% GVI were more than 50%. It indicated that the transitional district was the primary area, which should improve street vegetation and urban greening. In the modern district, the 33.7% sample points were lower than 25% GVI. They were concentrated in the western modern district, and they should be valued. (3) There were no low-GVI streets in the historic district. The GVI of only six sample points was lower than 15%. They were sample points 85, 125, 127, 132, 135, and 136. It accounted for only 4.3% of all sample points in the historic district. The sample points with GVI between 15% and 25% accounted for 11.5% of the historic district. In addition, there were 64 sample points with GVI higher than 50% in the three types of districts, accounting for 18.6% of the total sample points in Qingdao Coastal Streets. Among them, the historic district accounted for 68.75% of the 64 sample points, the modern district accounted for 18.75%, and the transitional district accounted for 12.5%. The highest GVI (81.1%) was sample point 287, located in the modern district.

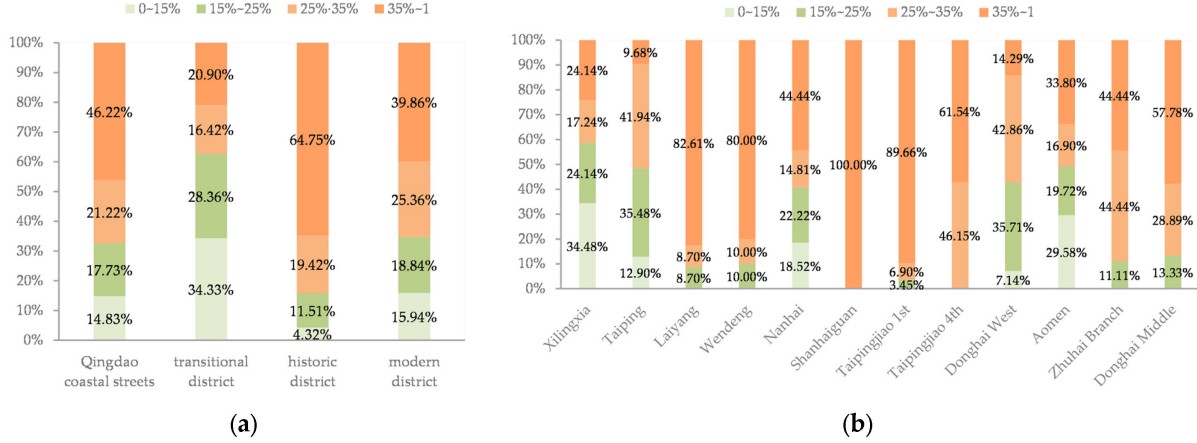

(**a**)                        (**b**)

**Figure 8.** (**a**) Proportions of the 4 GVI levels in the overall Qingdao Coastal Streets and the three types of districts; (**b**) proportions of the 4 GVI levels in the 12 roads.

From Figure 8b, we found that: (1) In Laiyang Road, Wendeng Road, Nanhai Road, Shanhaiguan Road, Taipingjiao 1st Road, Taipingjiao 4th Road, Aomen Road, Zhuhai Branch Road, and Donghai Middle Road, most of the sample points were the best green grade (GVI 35–100%), and the GVI even reached 100% in Shanhaiguan Road. In Taiping Road, Donghai West Road, and Zhuhai Branch Road, most of the sample points were the normal green grade (25–35%). In Xilingxia Road, the worst green grade (GVI 0–15%) accounted for the highest share. (2) In Xilingxia Road, Laiyang Road, Nanhai Road, Donghai West Road, and Aomen Road, the sample points with 0–25% GVI accounted for 40–60%. There was not enough green quantity in nearly 50% sample points of the five roads. And the pedestrians did not have good green perception.

### 3.2. The VICO Analysis

After standardizing the VICO data, the mean and standard deviation of the three district types and 12 roads were calculated (Figure 9). By comparing the average values (Figure 9a) of the three types of districts, it showed that VICO-historic (0.334) > VICO-modern (0.045) > VICO-transitional (−0.785). The VICO-historic was the best, which was far higher than the others. The second was the VICO-modern, which was close to the mean value of the overall coastal street. The VICO-transitional was a very poor score. In addition, the standard deviations of the three types of districts were high. It indicated that the fluctuation range of VICO was large.

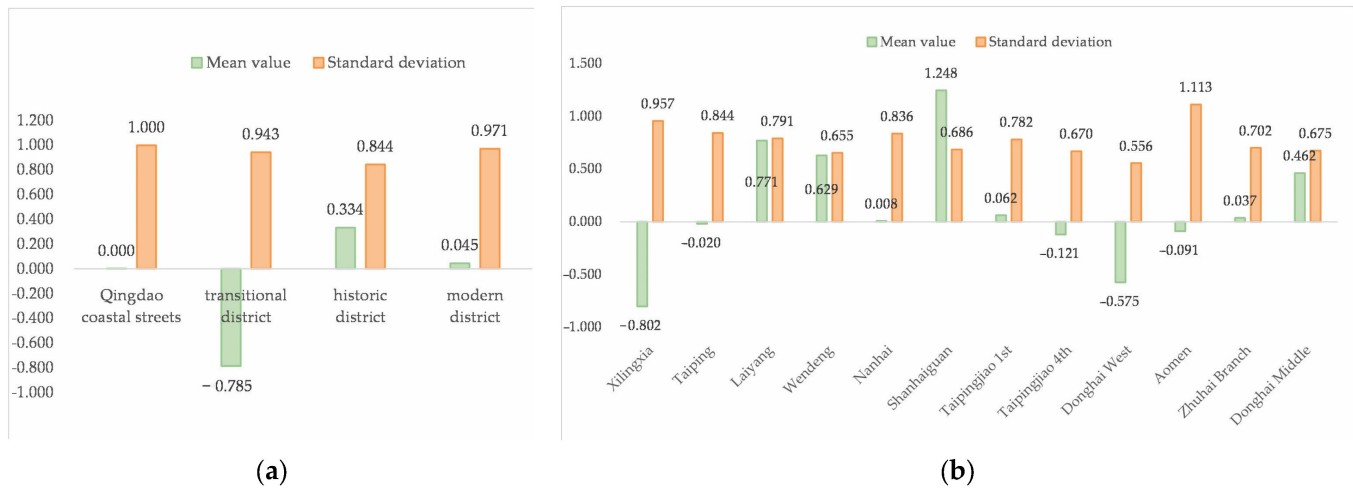

(**a**)　　　　　　　　　　　　　　　　　　　　　　(**b**)

**Figure 9.** (**a**) The descriptive analysis of VICO values in the overall Qingdao Coastal Streets and the three types of districts; (**b**) the descriptive analysis of VICO values in the 12 roads.

From Figure 9b, we found that: (1) The VICO was high in Laiyang Road (0.771), Wendeng Road (0.629), Shanhaiguan Road (1.248), and Donghai Middle Road (0.462). And it was the highest in Shanhaiguan Road. They were significantly better than the mean value in overall Qingdao Coastal Streets; Taiping Road (−0.020), Nanhai Road (0.008), Taipingjiao 1st Road (0.062), Taipingjiao 4th Road (−0.121), Aomen Road (−0.091), and Zhuhai Branch Road (0.037) were close to the mean value; Xilingxia Road (−0.802) and Donghai West Road (−0.575) were low VICO. (2) The standard deviation (1.113) was highest in Aomen Road, which indicated that the VICO distribution was more volatile. The standard deviations of the other 11 roads were in the range of 0.5 to 1.0.

According to the VICO result of the Z standard value, the top 115 sample points were classified as high VICO, the bottom 115 sample points were classified as low VICO, and the remaining 114 sample points were classified as medium VICO (Table 2).

**Table 2.** The VICO grades of sample points.

| Grade | High VICO | Medium VICO | Low VICO |
|---|---|---|---|
| Sample points | 17, 18, 19, 21, 22, 23, 25, 28, 36, 37, 68, 70, 78, 79, 90, 91, 92, 93, 94, 96, 97, 101, 104, 105, 106, 108, 109, 111, 112, 113, 114, 116, 117, 118, 121, 122, 140, 142, 144, 145, 146, 148, 149, 150, 151, 153, 154, 155, 156, 157, 158, 159, 160, 161, 162, 163, 166, 171, 177, 179, 180, 182, 183, 184, 185, 186, 187, 188, 189, 190, 193, 199, 201, 203, 204, 205, 206, 264, 266, 267, 269, 272, 273, 274, 275, 276, 279, 282, 284, 285, 286, 287, 288, 289, 290, 297, 300, 301, 306, 308, 311, 318, 322, 323, 325, 327, 328, 331, 332, 334, 335, 339, 341, 342, 343 | 13, 14, 15, 16, 20, 26, 27, 31, 33, 34, 35, 65, 67, 69, 71, 72, 73, 74, 75, 76, 77, 80, 81, 82, 83, 84, 98, 100, 102, 103, 107, 110, 115, 119, 120, 123, 128, 130, 133, 137, 138, 139, 141, 143, 147, 152, 164, 167, 168, 170, 172, 173, 174, 178, 181, 191, 194, 195, 196, 198, 200, 202, 208, 209, 216, 218, 222, 234, 235, 236, 237, 238, 239, 240, 242, 243, 258, 259, 260, 261, 265, 271, 277, 278, 280, 281, 283, 291, 292, 293, 296, 298, 299, 302, 303, 304, 305, 307, 310, 313, 314, 316, 317, 319, 320, 321, 324, 326, 329, 333, 337, 338, 340, 344 | 1, 2, 3, 4, 5, 6, 7, 8, 9, 10, 11, 12, 24, 29, 30, 32, 38, 39, 40, 41, 42, 43, 44, 45, 46, 47, 48, 49, 50, 51, 52, 53, 54, 55, 56, 57, 58, 59, 60, 61, 62, 63, 64, 66, 85, 86, 87, 88, 89, 95, 99, 124, 125, 126, 127, 129, 131, 132, 134, 135, 136, 165, 169, 175, 176, 192, 197, 207, 210, 211, 212, 213, 214, 215, 217, 219, 220, 221, 223, 224, 225, 226, 227, 228, 229, 230, 231, 232, 233, 241, 244, 245, 246, 247, 248, 249, 250, 251, 252, 253, 254, 255, 256, 257, 262, 263, 268, 270, 294, 295, 309, 312, 315, 330, 336 |

Figure 10 shows the spatial distribution of VICO in Qingdao Coastal Streets. The high VICO was mainly distributed in the historic district and the eastern modern district, especially in Laiyang Road, Wendeng Road, Shanhaiguan Road, the eastern Aomen Road, and the eastern Shanghai Middle Road. The distribution of high VICO was consistent with the high GVI. The medium VICO had a scattered distribution in the historic district and modern district, especially in Taiping Road, Nanhai Road, the eastern Taipingjiao 1st Road, and Taipingjiao 4th Road. The low VICO was mainly distributed in the transitional district and the western modern district, especially in Xilingxia Road, Donghai west Road, and the western Aomen Road.

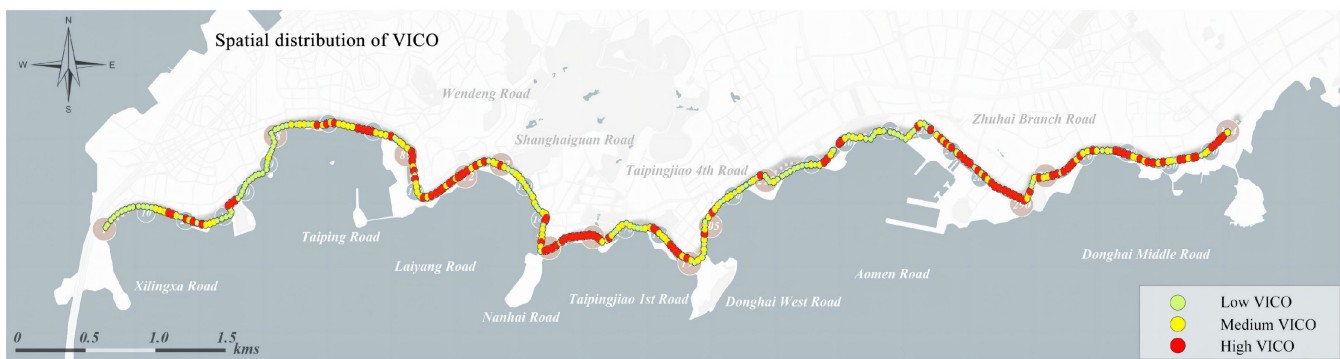

**Figure 10.** Spatial distribution of VICO.

The proportion of each VICO level is shown in Figure 11. As Figure 11a shows: in the transitional district, 65.7% of sample points had low VICO, and more than 50% of the sample points had low comfort perception for the public; in the historic district, 48.2% of the sample points had high VICO, nearly 50% of the street space was a comfortable walkable environment; in the modern district, medium VICO accounted for 37.7%, high VICO accounted for 27.5%, and low VICO accounted for 34.8%. The results revealed that the distribution of the three-level VICO was relatively balanced in the modern district.

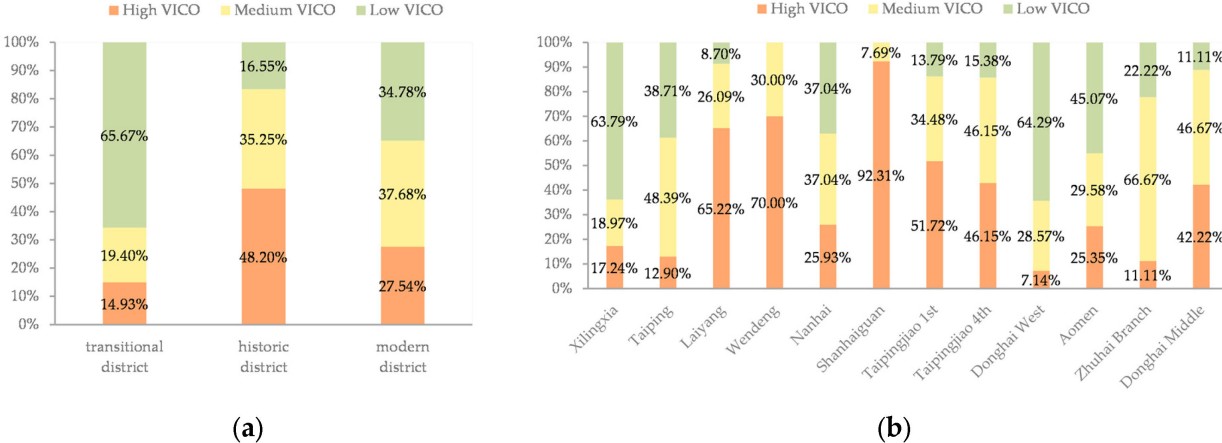

**Figure 11.** (**a**) Proportions of the 3 VICO levels in the three types of districts; (**b**) proportions of the 3 VICO levels in the 12 roads.

Figure 11b shows that: (1) In Laiyang Road, Wendeng Road, Shanhaiguan Road, Taipingjiao 1st Road, and Taipingjiao 4th Road, high VICO sample points had the highest proportion. And Shanhaiguan Road reached 92.31%. There were no low VICO points in Wendeng Road and Shanhaiguan Road. (2) In Taiping Road, Nanhai Road, Taipingjiao 4th Road, Zhuhai Branch Road, and Donghai Middle Road, medium VICO sample points had the highest proportion. (3) In Xilingxia Road, Nanhai Road, Donghai West Road, and Aomen Road, low VICO sample points had the highest proportion. Among them, the problems of street quality were the most significant in Xiling Xia Road and Dong Hai West Road. (4) The roads with the highest proportions of high VICO, medium VICO, and low VICO were Shanhaiguan Road, Zhu Hai Branch Road, and Dong Hai West Road, respectively.

### 3.3. Correlation Analysis of the VICO and GVI

This study conducted Pearson correlation analysis and regression analysis on the GVI and the VICO of the 344 sample points in Qingdao Coastal Streets by SPSS25.0 (Table 3).

**Table 3.** Correlation analysis of the VICO and GVI.

| VICO | Overall GVI | | 0–35% GVI | | >35% GVI |
|---|---|---|---|---|---|
| Pearson correlation | 0.769 ** | | 0.717 ** | | 0.390 ** |
| **VICO** | | **0~15% GVI** | **15~25% GVI** | **25~35% GVI** | |
| Pearson correlation | | 0.641 ** | 0.468 ** | 0.400 ** | |

** Significant at the 0.01 level.

In Table 3, we found that the correlation analysis for all sample points and the four GVI grades sample points, the GVI and VICO in Qingdao Coastal Streets were both positively correlated. This result was consistent with the GVI study of Xiaoxi. In the paper, 35% was taken as the threshold for GVI analysis. The results showed that: when the GVI was from 0~35%, the correlation between the GVI and VICO was a strong positive correlation (0.717); when the GVI was higher than 35%, the correlation between them was a weak positive correlation (0.390). The GVI of 0~35% was further divided into three grades: 0~15%, 15~25%, and 25~35%, and analyzed. Table 3 shows that: with the growth of GVI, the positive correlation between GVI and VICO gradually decreased, from 0.670 to 0.405. It indicated that the growth of street GVI can make the VICO increase faster in low GVI streets, and with the increase in street GVI, the improvement efficiency of street VICO gradually slowed and no longer became significant. In addition, the red

lines (Figure 12) in the scatter plot showed the relationship between the GVI and the VICO: Y(VICO) = 1.184ln(X(GVI)) + 4.376. $R^2$ (0.606) meant a good fit between them.

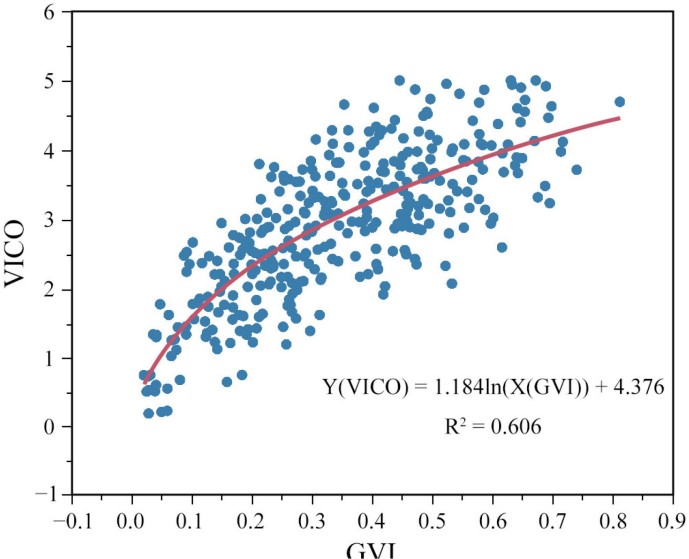

**Figure 12.** Scatter diagram of GVI and VICO.

## 4. Discussion

### *4.1. The GVI of Qingdao Coastal Streets*

By analyzing the Street View images in three types of districts and 12 sections in Qingdao Coastal Streets, it can be seen that the street GVI was influenced by vegetation canopy size, street width, and street function. The lush roadside vegetation and the small streets with human scale could improve the GVI and create high-quality green space [85,125]. In the eastern modern district, the green plants were lush, and the GVI was generally higher than 25%. A large number of street trees, shrubs, and lawns were planted on both sides of the streets. Smaller canopies and wider streets caused more plant canopy gaps and discontinuous interfaces that reduced the GVI. It was consistent with the viewpoint of Wang et al. (2022) [43]. The GVI of sample points 38–52 was lower than 15%. It had a poor green perception grade. The pedestrian view was sheltered by the fences on the north side of the streets; only a few patches of vegetation could be seen. The plants behind the fence were not tall and sparse. And the green space was also lacking on the south side of the streets.

Meanwhile, the tourist and living streets were quite demanding for street green quantity in order to improve the safety and pleasure perception. It was consistent with the studies of Araya et al. (2007) [126], Xiao et al. (2021) [79], and Bal et al. (2009) [127]. The GVI of sample points 13–37 was generally higher than 35%. The blocks were in a residential area with a small-scale street. The pedestrian and vehicle traffic was separated by plants. It could improve pedestrian safety and landscape quality in the streets. Commercial and coastal streets had a high vehicle occurrence rate and a low GVI [119,128]. The GVI of sample points 57–60 was very low (<0.05). The railway station, bus station, and many commercial buildings were around. The pedestrian and vehicular traffic was heavy. There were almost no green spaces. In addition, in order to improve vehicular traffic, the streets were made wide, which squeezed the urban green space. There were few plants on both sides of the streets where the sample points 224–228 and 246–249 were located, and almost none were planted on the side close to the sea. They provided the buildings and pavements with direct views of the sea but made the GVI very low. In addition, the GVI was low in the typical modern streets. They were concentrated in the western modern district. The features of these streets were: wide roads, open spaces, and high buildings. Although there

was some lawn or bushes, they accounted for the low proportion in the pedestrian view. That caused the low GVI.

### 4.2. The VICO of Qingdao Coastal Streets

Based on the VICO result and street view images, it can be seen that the street VICO was influenced by the GVI, plant richness, street scale, and landscape diversity.

The vegetation, which had dense coverage, high canopy density, and rich layering, brought a high VICO [129]. The plants were along the street and sidewalk, and the layers and contours of the vegetation communities were in accordance with the rules of beauty (sample points 23 and 26). In addition, the basic function of the street tree was to provide a clear physical separation and demarcation between the coastal trails and vehicular traffic [72,130]. It could ensure pedestrian safety, comfort, and sea views (sample points 17, 19, and 22). The planting could change the perceived width of the roads by improving the plant canopy density. And these streets had a human scale and a sense of security (sample points 101–119). The fence separated the plants from the pavement on the side away from the sea in the streets where sample points 2, 44~47, etc., were located, and the plants had poor permeability and were a single species. There was no verge planting zone adjacent to the pavement, which impacted pedestrians' perception of naturalness. The buildings that had historical styles and features with culture and pleasant colors had high VICO [131]. Three blocks (sample points 91–97, 145–163, and 177–190) had lush green vegetation, a suitable human scale, and faintly visible historic buildings. The quiet and harmonious atmosphere was the main reason for the extremely high VICO.

In addition, the street scale at high VICO sample points was comfortable, and the high-rise building impacted pedestrian spatial perception [132]. The high-rise buildings were dense and caused a sense of depression (sample points 58–61 and 216–218). The western modern district (sample points 207–257) was located at the fringe of the old and new urban areas in Qingdao. The street district had poor green coverage, wide vehicular lanes, a lot of vehicles, and many high-rise buildings, which led to the low VICO in the western modern district. It was confirmed that the iconic landscape and the landscape furniture not only increased public activities but also affected emotions [133]. There were a few beautiful plant landscapes and attractive iconic buildings at the seaside (sample points 1, 32, and 135). The pavement was shabby and lacked street furniture (sample points 46 and 49). The fence separated the coastal trails from the sea; it impacted the pedestrian perception of the coast (sample points 9 and 55).

### 4.3. Comparative Analysis of GVI and VICO

Pearson correlation analysis was performed between GVI and VICO of Qingdao Coastal Streets in the study. The results showed that they had a strong positive correlation, and their distribution and fluctuation were also highly consistent in Qingdao Coastal Streets. The study indicated that when the street GVI was lower than 25%, the GVI had a significant impact on public visual perception. With the increase in GVI, the influence of GVI on VICO gradually reduced. It is consistent with the results of Tan Shaohua (2016) [134] and Wang Yangyang (2021) [5]. Therefore, the low-GVI urban blocks were the primary areas for urban regeneration and greening quality improvement. As the reference factor for street landscape design, GVI is used to optimize the street space. It contributes to improving the public's visual satisfaction reasonably, accurately, and rapidly.

### 4.4. Limitations and Future Work

This study still has some limitations: The BSV images of Qingdao Coastal Streets were collected in summer, but some sections had plants with seasonal changes. In future studies, we will collect representative samples from the four seasons to conduct all-season research. The overall greening quality was high, and the landscape was attractive and distinctive in Qingdao Coastal Streets. Considering the types of coastal and urban environments, the regression equation may not be fully applicable to all coastal streets; VICO evaluation can

be combined with other methods, such as recording video, VR panorama, point cloud, etc. It would make the evaluation result more accurate and precise. Except for GVI and VICO, more factors should be added, and a comprehensive study will be conducted in future research, such as microclimate, noise interference, and the willingness to walk. That would enhance the greening design and improve the public's health in urban coastal streets.

## 5. Conclusions

This study analyzed the spatial distribution and influencing mechanism of GVI and VICO in Qingdao Coastal Streets. The result showed that: (1) the green quantity of the overall coastal streets was high, with an average GVI of 0.344. Meanwhile, the distribution of GVI in the three types of districts was relatively balanced. The street greening was lush in the historic district, and attention should be given to street vegetation and urban greening in the transitional district and the western modern district. (2) The spatial distribution of GVI and VICO was very similar. The VICO should be improved in Xiling Xia Road and Dong Hai West Road. (3) There was a significant positive correlation between GVI and VICO in Qingdao Coastal Streets. Along with the increase in GVI value, the acceleration of the VICO increase was decreasing. And their influence mechanism was the logarithmic equation. In streets with low greenery, by increasing the vertical interface area of the street vegetation, the public's VICO can be significantly improved, which is beneficial to the public's mental health. In the high GVI streets, the forms, types, and colors of vegetation should be improved. And in some street blocks, there was not enough ground space for vegetation greening; therefore, vertical greening was necessary, such as climbing plants, green walls, and furniture greening. This was a quantitative study on urban greening in coastal streets, and it revealed the relationship and influence mechanism between the important physical feature (GVI) and public psychological perception (VICO). It could not only improve the visual quality of Qingdao Coastal Streets, but also provide an important theoretical and application basis for the greening design and construction of urban streets.

**Author Contributions:** Conceptualization, D.S., X.J., W.G. and M.L.; methodology, D.S., X.J. and W.G.; software, D.S., X.J. and F.Z.; validation, D.S., W.G. and M.L.; formal analysis, D.S., W.G. and M.L.; investigation, D.S., X.J., Y.Y., Y.M., M.Y. and J.L.; resources, D.S., F.Z., Y.Y. and Y.M.; data curation, D.S. and X.J.; writing—original draft preparation, D.S., X.J. and Y.M.; writing—review and editing, D.S., X.J., Y.Y. and F.Z.; visualization, X.J., F.Z., M.Y. and J.L.; supervision, D.S., W.G. and M.L.; project administration, D.S., W.G. and M.L.; funding acquisition, D.S. and W.G. All authors have read and agreed to the published version of the manuscript.

**Funding:** The authors acknowledge the support by the Project of Liaoning Provincial Department of Education (2021) NO. 254.

**Institutional Review Board Statement:** Not applicable.

**Informed Consent Statement:** Not applicable.

**Data Availability Statement:** Interview transcripts and interpreted statements supporting this study's findings and the smart contract codes are available from the corresponding author upon reasonable request.

**Acknowledgments:** We would like to thank the journal experts who edited this paper. We also appreciate the constructive suggestions and comments on the manuscript from the reviewers and editors.

**Conflicts of Interest:** The authors declare no conflict of interest.

## Appendix A

**Table A1.** GVI and VICO values of 344 sample points.

| NUM | VICO | GVI | NUM | VICO | GVI |
|---|---|---|---|---|---|
| 001 | 0.214 | 0.059 | 050 | 1.212 | 0.138 |
| 002 | 1.840 | 0.113 | 051 | 2.933 | 0.149 |
| 003 | 0.661 | 0.080 | 052 | 0.538 | 0.059 |
| 004 | 1.763 | 0.101 | 053 | 1.604 | 0.312 |
| 005 | 0.632 | 0.158 | 054 | 0.734 | 0.183 |
| 006 | 2.152 | 0.248 | 055 | 1.206 | 0.201 |
| 007 | 2.251 | 0.206 | 056 | 2.087 | 0.231 |
| 008 | 1.886 | 0.168 | 057 | 0.500 | 0.038 |
| 009 | 1.619 | 0.217 | 058 | 1.316 | 0.041 |
| 010 | 1.318 | 0.190 | 059 | 0.730 | 0.019 |
| 011 | 1.198 | 0.164 | 060 | 0.198 | 0.049 |
| 012 | 2.295 | 0.235 | 061 | 1.712 | 0.168 |
| 013 | 2.341 | 0.406 | 062 | 2.502 | 0.205 |
| 014 | 3.043 | 0.320 | 063 | 2.189 | 0.138 |
| 015 | 2.895 | 0.306 | 064 | 2.793 | 0.215 |
| 016 | 2.864 | 0.294 | 065 | 3.279 | 0.214 |
| 017 | 3.561 | 0.566 | 066 | 2.345 | 0.169 |
| 018 | 3.549 | 0.512 | 067 | 3.224 | 0.333 |
| 019 | 3.010 | 0.601 | 068 | 3.919 | 0.328 |
| 020 | 2.341 | 0.253 | 069 | 3.136 | 0.259 |
| 021 | 4.417 | 0.567 | 070 | 4.145 | 0.306 |
| 022 | 3.307 | 0.675 | 071 | 3.300 | 0.302 |
| 023 | 3.471 | 0.687 | 072 | 3.548 | 0.253 |
| 024 | 1.409 | 0.207 | 073 | 3.393 | 0.245 |
| 025 | 3.659 | 0.642 | 074 | 3.327 | 0.280 |
| 026 | 2.527 | 0.434 | 075 | 3.073 | 0.288 |
| 027 | 2.634 | 0.331 | 076 | 3.086 | 0.224 |
| 028 | 2.854 | 0.475 | 077 | 3.569 | 0.261 |
| 029 | 2.085 | 0.281 | 078 | 4.211 | 0.411 |
| 030 | 1.943 | 0.146 | 079 | 4.302 | 0.431 |
| 031 | 2.413 | 0.470 | 080 | 3.412 | 0.425 |
| 032 | 1.588 | 0.185 | 081 | 3.312 | 0.307 |
| 033 | 2.381 | 0.343 | 082 | 2.576 | 0.266 |
| 034 | 2.264 | 0.304 | 083 | 3.250 | 0.331 |
| 035 | 3.829 | 0.436 | 084 | 3.534 | 0.279 |
| 036 | 4.597 | 0.641 | 085 | 1.865 | 0.126 |
| 037 | 3.527 | 0.457 | 086 | 1.539 | 0.167 |
| 038 | 1.436 | 0.090 | 087 | 2.646 | 0.195 |
| 039 | 1.099 | 0.074 | 088 | 2.473 | 0.213 |
| 040 | 1.107 | 0.142 | 089 | 2.526 | 0.199 |
| 041 | 2.014 | 0.149 | 090 | 4.329 | 0.406 |
| 042 | 1.604 | 0.178 | 091 | 4.869 | 0.471 |
| 043 | 1.915 | 0.267 | 092 | 4.516 | 0.491 |
| 044 | 1.722 | 0.127 | 093 | 3.181 | 0.476 |
| 045 | 1.508 | 0.121 | 094 | 4.729 | 0.496 |
| 046 | 1.644 | 0.144 | 095 | 1.819 | 0.207 |
| 047 | 1.287 | 0.123 | 096 | 3.847 | 0.551 |
| 048 | 0.174 | 0.028 | 097 | 3.050 | 0.477 |
| 049 | 1.006 | 0.065 | 098 | 3.387 | 0.390 |

**Table A1.** *Cont.*

| NUM | VICO | GVI | NUM | VICO | GVI |
|-----|------|-----|-----|------|-----|
| 099 | 2.551 | 0.184 | 149 | 4.719 | 0.654 |
| 100 | 3.070 | 0.320 | 150 | 4.085 | 0.580 |
| 101 | 3.581 | 0.489 | 151 | 3.769 | 0.637 |
| 102 | 2.817 | 0.387 | 152 | 2.263 | 0.409 |
| 103 | 3.461 | 0.435 | 153 | 4.546 | 0.653 |
| 104 | 4.161 | 0.451 | 154 | 4.896 | 0.647 |
| 105 | 4.278 | 0.360 | 155 | 4.937 | 0.632 |
| 106 | 3.673 | 0.420 | 156 | 3.887 | 0.640 |
| 107 | 3.400 | 0.300 | 157 | 4.073 | 0.625 |
| 108 | 3.685 | 0.415 | 158 | 3.943 | 0.616 |
| 109 | 4.425 | 0.437 | 159 | 5.000 | 0.630 |
| 110 | 2.942 | 0.406 | 160 | 3.711 | 0.583 |
| 111 | 4.033 | 0.361 | 161 | 3.706 | 0.740 |
| 112 | 4.260 | 0.395 | 162 | 3.874 | 0.649 |
| 113 | 4.651 | 0.353 | 163 | 4.124 | 0.670 |
| 114 | 4.951 | 0.523 | 164 | 2.582 | 0.615 |
| 115 | 3.328 | 0.355 | 165 | 1.901 | 0.418 |
| 116 | 3.828 | 0.556 | 166 | 4.461 | 0.693 |
| 117 | 3.127 | 0.587 | 167 | 2.480 | 0.456 |
| 118 | 4.064 | 0.583 | 168 | 2.057 | 0.532 |
| 119 | 2.413 | 0.333 | 169 | 2.112 | 0.314 |
| 120 | 3.005 | 0.239 | 170 | 2.956 | 0.366 |
| 121 | 3.066 | 0.447 | 171 | 2.927 | 0.597 |
| 122 | 3.606 | 0.496 | 172 | 2.315 | 0.521 |
| 123 | 2.542 | 0.454 | 173 | 2.785 | 0.527 |
| 124 | 2.399 | 0.231 | 174 | 2.193 | 0.393 |
| 125 | 1.543 | 0.103 | 175 | 1.394 | 0.193 |
| 126 | 2.473 | 0.180 | 176 | 1.964 | 0.275 |
| 127 | 2.456 | 0.128 | 177 | 3.386 | 0.558 |
| 128 | 2.756 | 0.290 | 178 | 2.834 | 0.481 |
| 129 | 1.177 | 0.257 | 179 | 4.110 | 0.716 |
| 130 | 2.950 | 0.272 | 180 | 3.969 | 0.713 |
| 131 | 1.372 | 0.180 | 181 | 2.560 | 0.406 |
| 132 | 1.783 | 0.109 | 182 | 3.143 | 0.512 |
| 133 | 2.992 | 0.208 | 183 | 2.924 | 0.494 |
| 134 | 2.483 | 0.217 | 184 | 3.281 | 0.577 |
| 135 | 2.230 | 0.091 | 185 | 3.821 | 0.576 |
| 136 | 2.339 | 0.096 | 186 | 3.673 | 0.618 |
| 137 | 2.873 | 0.226 | 187 | 4.223 | 0.578 |
| 138 | 2.944 | 0.384 | 188 | 4.058 | 0.598 |
| 139 | 3.426 | 0.397 | 189 | 3.223 | 0.695 |
| 140 | 3.702 | 0.483 | 190 | 3.659 | 0.503 |
| 141 | 2.019 | 0.421 | 191 | 2.873 | 0.462 |
| 142 | 3.196 | 0.457 | 192 | 1.551 | 0.273 |
| 143 | 3.071 | 0.343 | 193 | 3.092 | 0.529 |
| 144 | 2.959 | 0.457 | 194 | 2.867 | 0.552 |
| 145 | 4.679 | 0.577 | 195 | 2.449 | 0.286 |
| 146 | 3.082 | 0.457 | 196 | 2.787 | 0.358 |
| 147 | 2.453 | 0.372 | 197 | 1.827 | 0.261 |
| 148 | 4.396 | 0.646 | 198 | 2.899 | 0.344 |

**Table A1.** *Cont.*

| NUM | VICO | GVI | NUM | VICO | GVI |
|---|---|---|---|---|---|
| 199 | 3.582 | 0.484 | 249 | 1.250 | 0.071 |
| 200 | 2.848 | 0.254 | 250 | 2.723 | 0.178 |
| 201 | 3.262 | 0.526 | 251 | 1.320 | 0.090 |
| 202 | 2.144 | 0.315 | 252 | 2.028 | 0.214 |
| 203 | 3.152 | 0.561 | 253 | 2.318 | 0.221 |
| 204 | 4.369 | 0.608 | 254 | 1.426 | 0.076 |
| 205 | 3.396 | 0.541 | 255 | 2.387 | 0.142 |
| 206 | 3.211 | 0.492 | 256 | 2.354 | 0.117 |
| 207 | 1.340 | 0.125 | 257 | 2.668 | 0.189 |
| 208 | 2.066 | 0.266 | 258 | 2.344 | 0.294 |
| 209 | 2.680 | 0.260 | 259 | 2.005 | 0.248 |
| 210 | 2.535 | 0.234 | 260 | 3.710 | 0.284 |
| 211 | 1.747 | 0.153 | 261 | 3.136 | 0.322 |
| 212 | 2.798 | 0.196 | 262 | 2.279 | 0.231 |
| 213 | 2.339 | 0.235 | 263 | 2.195 | 0.170 |
| 214 | 1.371 | 0.296 | 264 | 2.955 | 0.453 |
| 215 | 1.739 | 0.259 | 265 | 3.544 | 0.393 |
| 216 | 2.573 | 0.251 | 266 | 3.143 | 0.459 |
| 217 | 2.009 | 0.193 | 267 | 4.602 | 0.402 |
| 218 | 3.796 | 0.418 | 268 | 2.516 | 0.091 |
| 219 | 1.750 | 0.264 | 269 | 5.000 | 0.445 |
| 220 | 2.196 | 0.146 | 270 | 2.993 | 0.188 |
| 221 | 1.432 | 0.251 | 271 | 3.788 | 0.212 |
| 222 | 3.228 | 0.294 | 272 | 3.403 | 0.451 |
| 223 | 1.387 | 0.134 | 273 | 3.713 | 0.474 |
| 224 | 0.494 | 0.024 | 274 | 4.015 | 0.499 |
| 225 | 0.590 | 0.040 | 275 | 4.076 | 0.553 |
| 226 | 1.759 | 0.047 | 276 | 4.136 | 0.508 |
| 227 | 1.237 | 0.066 | 277 | 2.336 | 0.473 |
| 228 | 0.734 | 0.030 | 278 | 3.618 | 0.446 |
| 229 | 0.718 | 0.027 | 279 | 3.226 | 0.463 |
| 230 | 0.513 | 0.028 | 280 | 3.603 | 0.232 |
| 231 | 2.653 | 0.101 | 281 | 2.890 | 0.449 |
| 232 | 1.604 | 0.061 | 282 | 3.970 | 0.474 |
| 233 | 2.581 | 0.161 | 283 | 3.610 | 0.305 |
| 234 | 3.621 | 0.252 | 284 | 4.915 | 0.688 |
| 235 | 3.743 | 0.237 | 285 | 5.000 | 0.672 |
| 236 | 3.245 | 0.321 | 286 | 4.626 | 0.698 |
| 237 | 2.548 | 0.309 | 287 | 4.690 | 0.811 |
| 238 | 2.092 | 0.285 | 288 | 5.000 | 0.630 |
| 239 | 3.423 | 0.423 | 289 | 4.864 | 0.586 |
| 240 | 3.832 | 0.297 | 290 | 4.808 | 0.545 |
| 241 | 2.762 | 0.168 | 291 | 2.589 | 0.320 |
| 242 | 2.449 | 0.273 | 292 | 3.109 | 0.391 |
| 243 | 2.849 | 0.474 | 293 | 3.619 | 0.329 |
| 244 | 2.333 | 0.199 | 294 | 2.045 | 0.228 |
| 245 | 2.460 | 0.087 | 295 | 1.645 | 0.266 |
| 246 | 1.283 | 0.040 | 296 | 2.891 | 0.261 |
| 247 | 1.322 | 0.035 | 297 | 4.113 | 0.403 |
| 248 | 1.295 | 0.040 | 298 | 2.869 | 0.400 |

**Table A1.** *Cont.*

| NUM | VICO | GVI | NUM | VICO | GVI |
|---|---|---|---|---|---|
| 299 | 3.217 | 0.385 | 322 | 3.959 | 0.385 |
| 300 | 4.066 | 0.397 | 323 | 3.321 | 0.459 |
| 301 | 4.276 | 0.415 | 324 | 3.765 | 0.388 |
| 302 | 2.155 | 0.379 | 325 | 3.500 | 0.533 |
| 303 | 3.497 | 0.375 | 326 | 2.859 | 0.437 |
| 304 | 3.532 | 0.273 | 327 | 3.966 | 0.531 |
| 305 | 3.710 | 0.299 | 328 | 3.278 | 0.485 |
| 306 | 3.809 | 0.340 | 329 | 3.016 | 0.390 |
| 307 | 3.060 | 0.330 | 330 | 1.575 | 0.211 |
| 308 | 4.081 | 0.334 | 331 | 3.746 | 0.480 |
| 309 | 2.491 | 0.221 | 332 | 4.015 | 0.533 |
| 310 | 3.313 | 0.311 | 333 | 2.949 | 0.378 |
| 311 | 4.279 | 0.333 | 334 | 3.906 | 0.450 |
| 312 | 2.842 | 0.219 | 335 | 4.488 | 0.485 |
| 313 | 2.155 | 0.259 | 336 | 2.092 | 0.154 |
| 314 | 3.359 | 0.352 | 337 | 2.809 | 0.299 |
| 315 | 2.196 | 0.200 | 338 | 2.925 | 0.442 |
| 316 | 3.773 | 0.446 | 339 | 4.261 | 0.477 |
| 317 | 3.513 | 0.350 | 340 | 3.811 | 0.345 |
| 318 | 4.213 | 0.495 | 341 | 3.860 | 0.496 |
| 319 | 3.516 | 0.409 | 342 | 4.544 | 0.490 |
| 320 | 2.495 | 0.289 | 343 | 3.965 | 0.459 |
| 321 | 3.096 | 0.286 | 344 | 2.878 | 0.228 |

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
