# Peer review of "The Relation between Green Visual Index and Visual Comfort in Qingdao Coastal Streets"

_buildings, doi:10.3390/buildings13020457_

Round 1

Reviewer 1 Report

Overall, this study is based on good science and aims to answer some interesting and important research questions.  This manuscript is very interesting for publication. I have some recommendations that should be corrected before publication. 

Major concerns:

1. Abstract_ all showed common knowledge, some interesting and quantify results should be concluded. There is still a lack of brief significance and conclusions. How is the relationship and mechanism of the influence of Green View Index (GVI) and Visual Comfort (VICO) on mental health confirmed?

2. Introduction_ The literature reviews were not updated, many related studied were not concerned. A comprehensive literature review should be added to clearly reflect: 1) what the relevant research progress is and, 2) why your proposal is important. 3) What are the research purposes and questions?

3. Methods_ A flow chart to describe the procedure of the experiment is needed. 

4. The discussion in this paper is incomplete. Discussion is an extension of the research results. Discussions should be based on the scientific nature and rationality of the results, combined with literature for in-depth analysis, and in-depth analysis of the mechanism of the results, the similarities and differences between the analysis and previous results, and attention should be paid to whether the comparison is consistent with similar reports. And explain why the result occurs and what the result means. There is a distinct lack of discussion regarding previous literature's results. Therefore, the authors need more comparisons with previous studies. 

5. In addition, the limitations of this study should be supplemented. Future directions should be discussed. I recommend that you highlight the challenges and limitations of the current study to ensure that the results are interpreted correctly in these contexts - you have already cited some literature. Still, there are many more that explore this further. Does the method improve accuracy? Confirmed the previous research?  I think it is important to add these elements.

6. The precise quantitative results of previous literature need to be mentioned a bit more so that readers can know if your research is consistent with previous studies. Finding studies that measured your results, in the same way, would also help to see differences between your research and others. In addition, I suggest you emphasize the contribution of research.

7.Conclusions_ Conclusions were all common knowledge. How to apply the results to real the greening design and construction of urban streets? How to improve human well-being and health?Some prospective statements should be highlighted. What is the impact relationship? What is the impact mechanism? Where is the specific statement?

Minor comments:

1. There are still some problems with language expression in the manuscript, and the author is suggested to revise the language style further.

2. Figures 3, 4 and 5 need to be updated. The resolution is not enough, and the text in the picture cannot be seen clearly.

3. Figure 1.  Study area and distribution of sampling points need to be re expressed. It is too vague whether provinces or cities or countries.

4. The correlation analysis in Table 3 is incorrect.

Reviewer 2 Report

Theoretical foundation needs to be.addressed using literature review section.

It should address both green visual index and visual comfort.

That has to include the definition and historic flow of research.related to the concept.

Also, authors has to present the implication of this work.

Conclusion should be strengthened substantially.

Reviewer 3 Report

This paper explores the relationship between Green Visual Index and Visual Comfort in Qingdao Coastal Streets. The topic is timing, and the paper is well-written. The methodology has been clearly explained. The results are based on evidence and have also clearly been discussed. However, the discussion and introduction can be improved. Based on the above comments, the manuscript can be accepted after revision. The following are specific comments:

In the abstract, “Green View Index (GVI) can reflect the three-dimensional green quantity”. Please carefully use this claim, as the GVI is based on images which may not be three-dimensional.

Line 52, please use the full name at the first mention.

The logic of the introduction should be improved. For example, Line 51-55, “The most critical issues in the study of Qingdao coastal street are as follows: (1) Efficiently and accurately quantify analysis of the GVI and VICO in coastal streets. (2) Comparative analysis of the spatial heterogeneity of the GVI and VICO in various types of districts. (3) Study the relationship and influence mechanism between the GVI and VICO.” This paragraph suddenly appears without any background of coastal streets in Qingdao or other cities.

The introduction lacks the background of VICO. Also, visual comfort is subjective and elusive, and its definition varies across different disciplines. The following references have explained the definitions

https://www.sciencedirect.com/science/article/pii/S1470160X2200646X; https://www.sciencedirect.com/science/article/pii/S0169204622002341; Please clearly define visual comfort in this study.

The methodology is clear. I suggest adding a framework or diagram at the beginning to help readers understand the method easily.

The results of GVI and VICO are supported by evidence and have been clearly discussed. However, there is no comparison or discussion in Section 3.3.

I recommend adding limitations and future work at the Conclusion.

Reviewer 4 Report

The results obtained are well-grounded and convincing. The research material is well-structured, systematized, logically, and coherently presented. The title of the article reflects the content of the research. The abstract and keywords are appropriate. The introduction encompasses the description of the research relevance, previous research in this field, and the tasks of the research. The main text represents the research object and methods, results, and their evaluation. Conclusions summarize the main findings of the research. A list of references is appropriate.

The main remarks are the following:  

-        It is necessary to explain in the text the meaning of numbers in brackets, e.g., in line 227 it is not clear what is 23 and 36. Are they numbers of the evaluated images?

-        Perspectives of the method development could be discussed in the conclusions because visual comfort also can be evaluated using other methods (e.g., method of video-ecology developed by V. A. Filin where the main criteria of evaluation of visual environment ecological potential are homogeneity/heterogeneity, aggressiveness, and comfortability (Filin, 2001)).

- The English language of the article needs minor corrections in some places, e.g., in line 299 “the blocks was…”, in line 334 “Table 3 show…”, etc.

Round 2

Reviewer 1 Report

 I recommend accepting this manuscript for publication.

Author Response

Dear reviewer:

We sincerely thank you for your constructive and insightful comments on our previous submission, and thank you for your support in the publication of this manuscript. These comments are valuable and very helpful in revising and improving our paper, as well as they have important guiding significance for our research. In addition, we re-examined the manuscript to avoid small errors such as grammar, the information in the Figures and Tables. They were marked in yellow.                  

Best regard,  

The Authors

Reviewer 2 Report

It is well done

Author Response

Dear reviewer:

We sincerely thank you for your constructive and insightful comments on our previous submission. These comments are valuable and very helpful in revising and improving our paper, as well as they have important guiding significance for our research. In addition, we re-examined the manuscript to avoid small errors such as grammar, the information in the Figures and Tables. They were marked in yellow.                   

Best regard,  

The Authors

Reviewer 3 Report

The revision has addressed my main concerns. I only have one more suggestion. 

Please add one or two sentences to summarise the literature or present the research gap in Line 151.  This will highlight the contribution of this study.

After this revision, I think the manuscript can be accepted.

Author Response

Dear reviewer:

Thank you for your suggestion. To highlight the contribution of this study, we summarized the literature and presented the research gap (See line 152-154). It was marked in blue in the manuscript. In addition, we re-examined the manuscript to avoid small errors such as grammar, the information in the Figures and Tables. They were marked in yellow. 

Finally, we sincerely thank you for your constructive and insightful comments on our previous submission, and thank you for your support in the publication of this manuscript. These comments are valuable and very helpful in revising and improving our paper, as well as they have important guiding significance for our research.               

Best regard,  

The Authors